# Applicability of Statins in Metabolic Dysfunction-Associated Steatotic Liver Disease (MASLD)

**Thaninee Prasoppokakorn** [1,2]

1   Division of Gastroenterology, Department of Medicine, Faculty of Medicine, Chulalongkorn University and King Chulalongkorn Memorial Hospital, Thai Red Cross Society, Bangkok 10330, Thailand; thanineeeve@gmail.com or thaninee.p@chula.ac.th; Tel.: +66-2-256-4265; Fax: +66-2-256-4356
2   Department of Medicine, Queen Savang Vadhana Memorial Hospital, Chonburi 20110, Thailand

**Abstract:** Metabolic dysfunction-associated steatotic liver disease (MASLD) is the novel terminology encompassing liver disease associated with metabolic dysfunction, replacing the previous terminology of non-alcoholic fatty liver disease (NAFLD). This disease is strongly associated with metabolic disorders such as obesity, type 2 diabetes, and dyslipidemia. MASLD and dyslipidemia are deeply interconnected, driven by shared pathophysiological mechanisms. Emerging evidence suggests that statins, a class of lipid-lowering medications, may have beneficial effects on MASLD beyond their primary role in reducing cholesterol levels through several mechanisms, including anti-inflammatory, antioxidant, anti-fibrosis, and immunomodulatory effects. This review aims to summarize the efficacy of statins in the management of MASLD and provide insights into their potential mechanisms of action. It discusses the pathophysiology of MASLD and the role of statins in targeting key aspects of the disease. Additionally, the review examines the clinical evidence supporting the use of different statins in MASLD treatment and highlights their specific effects on liver enzymes, inflammation, and fibrosis. Furthermore, an algorithm for statin therapy in MASLD is proposed based on the current knowledge and available evidence.

**Keywords:** statin; steatotic liver disease; metabolic dysfunction; MASLD

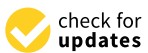

## 1. Introduction

Non-alcoholic fatty liver disease (NAFLD) is a global health concern, which affects approximately 25% of the population in Western countries [1] and 25–30% in the Asia-Pacific region [2]. In contrast, the term metabolic dysfunction-associated steatotic liver disease (MASLD) does not have a well-reported worldwide incidence [3]. MASLD is considered the hepatic expression of metabolic syndrome (MS) despite the possibility of genetic involvement and its occurrence in lean individuals without diabetes [4]. MASLD encompasses a diverse range of liver conditions characterized by the accumulation of fat in the liver. This heterogeneity is evident in the clinical and histologic spectrum of steatotic liver disease, ranging from isolated liver steatosis, referred to as simple steatosis, while others experience hepatocyte injury, ballooning, inflammation, and subsequent fibrosis, known as steatohepatitis (MASH) or the previous term of non-alcoholic steatohepatitis (NASH) [5].

Systemic insulin resistance (IR) plays a significant role in the development of hepatic steatosis in MASLD. Furthermore, the activation of the innate immune system and the lipotoxicity resulting from accumulated lipids are key factors driving the progression from simple steatosis to MASH. This theory proposes that a combination of insulin resistance, genetic and epigenetic factors, mitochondrial dysfunction, endoplasmic reticulum stress,

alterations in the microbiota, chronic inflammation, and the dysfunction of adipose tissue collectively contribute to the development and progression of MASH [4].

Early identification and management of MASLD are crucial to prevent disease progression and complications. Weight reduction is the most effective therapy for MASLD, as a 10% reduction can lead to the resolution of steatohepatitis and improve fibrosis by at least one stage. Moreover, it can reduce the risks of cardiovascular disease and diabetes [6]. Currently, there are no evidence-based drug therapies specifically recommended for the management of MASLD/MASH, highlighting a significant unmet clinical need [7]. However, in clinical practice, pharmacotherapy is commonly used to address the increased cardiovascular risk through anti-obesity, anti-diabetic, and lipid-lowering drugs [8]. In this article, we aim to review the utility of statin medications for MASLD treatment based on the underlying pathophysiology of the disease.

In previous studies, the evidence we reviewed primarily utilized the term NAFLD in clinical trials. Although MASLD and NAFLD are distinct terms, the new nomenclature of MASLD maintains a close connotation with NAFLD. The differences between MASLD and NAFLD are minimal, making it reasonable to assume that the findings from earlier NAFLD studies remain valid under the new MASLD definition [9,10]. In this literature review, we use the term NAFLD in alignment with the original references.

## 2. MASLD and Dyslipidemia

MASLD and dyslipidemia are deeply interconnected, driven by shared pathophysiological mechanisms. Dyslipidemia is one of the cardiometabolic criteria for MASLD, defined as triglycerides over 150 mg/dL or treatment with lipid-lowering drugs [3]. This condition is multifaceted and can be explained through several key mechanisms: **Firstly,** lipid metabolism: Dyslipidemia is characterized by increased hepatic production and decreased clearance of triglyceride-rich lipoproteins. These elevated levels of triglycerides can contribute to the development of hepatic steatosis in MASLD. **Secondly,** insulin resistance: IR promotes lipolysis in adipose tissue, leading to an increased influx of non-esterified fatty acids (NEFAs) into the liver and the subsequent accumulation of hepatic fat. These fatty acids are then taken up by the liver, promoting hepatic triglyceride accumulation and the development of MASLD [11,12]. **Thirdly,** inflammation and oxidative stress: the low-density lipoprotein cholesterol (LDL-C) and triglycerides elevation can trigger an inflammatory response in the liver, leading to the release of pro-inflammatory cytokines. This inflammation, along with oxidative stress, contributes to the progression from simple steatosis to steatohepatitis [13]. **Finally,** genetic and environmental factors: Genetic variations in the genes involved in lipid metabolism, insulin signaling, and inflammation can predispose individuals to dyslipidemia and MASLD. Additionally, environmental factors such as a sedentary lifestyle, unhealthy diet (high in saturated fats and refined carbohydrates), and obesity can exacerbate both conditions [14]. It is important to note that the relationship between dyslipidemia and MASLD is complex, with various factors contributing to their coexistence and progression. The proper management of dyslipidemia, including lifestyle modifications and pharmacotherapy if necessary, can help improve lipid profiles and potentially mitigate the progression of MASLD.

## 3. Statins

Statins are a selective, competitive inhibitor of hydroxymethylglutaryl-CoA (HMG-CoA) reductase, the enzyme responsible for converting HMG-CoA to mevalonate in the cholesterol synthesis pathway. The FDA-approved indications for statins are as follows: hyperlipidemia and mixed dyslipidemia, hypertriglyceridemia, atherosclerosis, primary prevention of ASCVD (atherosclerotic cardiovascular disease), and secondary prevention in

patients with clinical ASCVD [15]. HMG CoA reductase inhibitors have pleiotropic effects beyond their primary role. Statin inhibits the synthesis of isoprenoid intermediates that are necessary for activating certain intracellular and signaling proteins. As a result, statins exhibit anti-inflammatory, antioxidant, antiproliferative, and immunomodulatory effects. Furthermore, they contribute to plaque stability and prevent platelet aggregation. This pleiotropic effect is observed across all statins and is considered a class effect [16].

Table 1 presents the key pharmacokinetic characteristics of statins and summaries of the total available statins [17]. These medications can be classified into two groups based on their solubility. Hydrophilic compounds exhibit more significant active renal excretion, while lipophilic compounds are primarily eliminated by the liver. LDL-C lowering is <30%, 30–49%, and ≥50% in low, moderate, and high-intensity statin therapy, respectively [18].

**Table 1.** The pharmacokinetic characteristics of statins.

| | Simvastatin | Atorvastatin | Rosuvastatin | Pravastatin | Pitavastatin | Lovastatin | Fluvastatin |
|---|---|---|---|---|---|---|---|
| **Solubility** | Lipophilic | Lipophilic | Hydrophilic | Hydrophilic | Lipophilic | Lipophilic | Lipophilic |
| **Primary metabolic pathway** | CYP3A4 | CYP3A4 | CYP2C9-CYP2C19 | Glucuronidation-CYP3A4 | CYP2C9-CYP2C8 | CYP3A4 | CYP2C9 |
| **Bioavailability** | <5% | 12% | 20% | 18% | 80% | <5% | 10–35 |
| **Protein binding** | >95% | >98% | 88% | 43–54% | >95% | 96–98% | >98% |
| **Hepatic excretion** | 78–97% | >70% | 90% | 46–66% | 79% | >70% | >68% |
| **Renal excretion** | 13% | 2% | 10% | 60% | 15% | 30% | 6% |
| **Absorption** | 65–85% | 30% | 50% | 37% | 80% | 31% | 98% |
| **$T_{max}$,h** | 1.3–2.4 | 2–4 | 3–4 | 1–16 | 1 | 2–4 | 0.5–1.5 |
| **$T_{1/2}$** | 1.9–3 | 11–30 | 20 | 0.8–3 | 12 | 2.5–3 | 0.5–2.3 |
| **Low intensity** | 10 mg | - | - | 10–20 mg | 1 mg | 20 mg | 20–40 mg |
| **Moderate intensity** | 20–40 mg | 10–20 mg | 5–10 mg | 40–80 mg | 2–4 mg | 40 mg | 80 mg |
| **High intensity** | - | 40–80 mg | 20–40 mg | - | - | - | - |

## 4. Statin Mechanism of Actions and Pleiotropic Effects for MASLD

Due to the progression of MASLD and its disease spectrum, which includes comorbidities such as dyslipidemia and atherosclerotic heart disease, statins are undoubtedly beneficial for these conditions. Moreover, as fatty liver disease progresses through stages like simple steatosis, advanced fibrosis, cirrhosis, liver-related complications, and liver cancer, statins play diverse roles in each stage of disease progression, supported by the following research evidence (Figure 1).

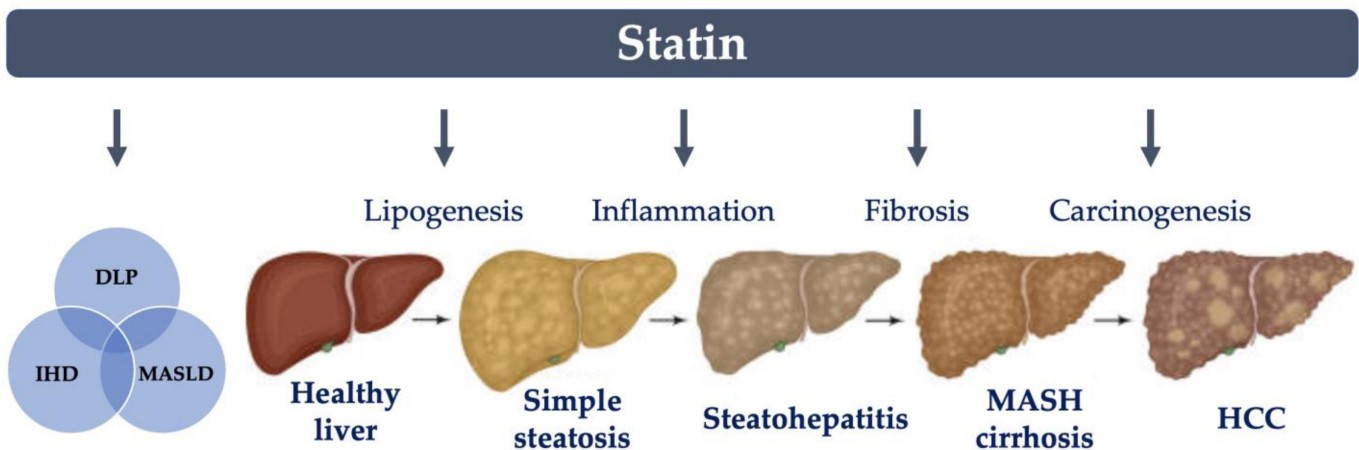

**Figure 1.** Statin mechanisms of action and pleiotropic effects for MASLD. DLP, dyslipidemia. IHD, ischemic heart disease. MASLD, metabolic dysfunction-associated steatotic liver disease. MASH, metabolic dysfunction-associated steatohepatitis. HCC, hepatocellular carcinoma.

*4.1. Lipogenesis*

Statin's effect on lipogenesis, specifically from a healthy liver to fatty liver disease, is a complex process that involves multiple factors. In a healthy liver, statins can help regulate lipogenesis by decreasing the production of cholesterol. As cholesterol levels decrease, it can lead to a decrease in the synthesis of fatty acids, triglycerides, and other lipids. Additionally, statins may have indirect effects on lipogenesis by improving insulin sensitivity and reducing insulin resistance, which can help regulate lipid metabolism.

In a steatotic liver, Paraoxonase 1 (PON1) is an enzyme primarily associated with high-density lipoprotein (HDL) particles and is involved in various physiological processes, including lipid metabolism and protection against oxidative stress. There is evidence to suggest that reduced levels or impaired activity of PON1 may contribute to the development and progression of liver steatosis. Statins decrease hepatic steatosis by reducing oxidative stress through increased activity of the antioxidant enzyme PON1 [19].

*4.2. Anti-Inflammation*

Chronic inflammation is a key component of steatosis, contributing to liver damage and disease progression. MASH progression is influenced by various factors, including free fatty acids (FFAs), inflammatory cytokines and adipokines, oxidative stress, and mitochondrial dysfunction [20]. The accumulation of FFA, particularly after beta- and omega-oxidation, leads to the generation of reactive oxygen species (ROS) [21]. Pro-inflammatory cytokines such as tumor necrosis factor-alpha (TNF-$\alpha$) and interleukin 6 (IL-6), which are produced by the liver and adipose tissue upon the activation of nuclear factor kappa B (NF-$\kappa$B), are typically elevated and also associated with IR.

Statins have been found to have various beneficial effects on inflammation. They can indirectly impact the availability of FFA for oxidation. By lowering cholesterol levels, statins may contribute to a decrease in FFA accumulation and the subsequent generation of ROS. Furthermore, statins decrease hepatic inflammation by suppressing the expression of pro-inflammatory cytokines such as TNF-$\alpha$ and IL-6, and transforming growth factor-beta 1 (TGF-$\beta$1) [22]. Moreover, statins increase the levels of PON1, an antioxidant and antiatherogenic enzyme primarily synthesized in the liver. PON1 has the ability to hydrolyze peroxides and lactones associated with lipoproteins, thereby reducing oxidative stress and inflammation. Additionally, it is associated with high-density lipoproteins (HDLs) in the bloodstream, resulting in an atheroprotective effect [19].

Beyond their hepatic effects, statins have also been shown to influence the activity of matrix metalloproteinases (MMPs), which play a critical role in vascular remodeling and atherogenesis. By inhibiting MMP activity, statins reduce vascular inflammation and the degradation of extracellular matrix components, contributing to the stabilization of atheromatic plaques and the prevention of aneurysm formation [23]. These effects underline the significant role of statins in reducing cardiovascular and vascular complications associated with MASLD.

Recent evidence suggests that statins may also mitigate inflammation through epigenetic mechanisms, particularly in conditions associated with early-life environmental insults such as undernutrition during pregnancy. These adverse conditions may induce epigenetic alterations, including DNA methylation and histone modifications, that predispose offspring to insulin resistance and a pro-inflammatory state [24]. By modulating inflammatory pathways and reducing oxidative stress, statins have the potential to counteract these epigenetic changes, thereby improving metabolic and inflammatory outcomes. Further research is needed to validate the role of statins in addressing inflammation related to epigenetic alterations.

Lastly, statins exert their pleiotropic effects by upregulating the gene expression of peroxisomal proliferator-activated receptor alpha (PPARα), a crucial regulator of fatty acid oxidation. This mechanism is particularly significant in improving peroxisomal and mitochondrial fatty acid oxidation, which is diminished in patients with MASH [25].

### 4.3. Anti-Fibrosis

The pathophysiology of fibrosis in fatty liver disease involves various mechanisms, including inflammation, oxidative stress, activation of hepatic stellate cells (HSCs), and deposition of extracellular matrix (ECM) components. Statins have been shown to exert anti-fibrotic effects in the context of fatty liver disease through several mechanisms. Statins have been found to inhibit the activation of HSCs, thereby reducing the production of ECM and limiting fibrosis progression [25]. Moreover, the anti-inflammatory effects of statins indirectly contribute to preventing the progression of liver fibrosis. By mitigating inflammation, statins help reduce the activation of HSCs and the subsequent fibrogenesis. Additionally, statins can interfere with pro-fibrotic signaling pathways that are involved in the development of fibrosis. For instance, they have demonstrated the ability to inhibit the TGF-β signaling pathway, which is a major driver of fibrosis. By inhibiting TGF-β signaling, statins can attenuate fibrogenesis and the deposition of extracellular matrix (ECM) [26]. Moreover, statins have been shown to utilize anti-fibrotic effects and reduce portal pressure by improving the function of liver sinusoidal endothelial cells (LSECs) through enhancing nitric oxide (NO) signaling. In MASH, the dysfunction of LSECs is supposed to occur prior to the development of portal hypertension and the subsequent fibrosis [27].

### 4.4. Anti-Carcinogenesis

Statins have demonstrated their ability to reduce the expression of pro-inflammatory and pro-fibrogenic mediators. In a preclinical model of hepatocellular carcinoma (HCC) associated with steatohepatitis and a high-fat and high-cholesterol diet, there were significant reductions in the expression of various inflammatory markers, including TNF-α, IL-6, IL-1β, interferon (IFN)-γ, and TGF-β1. Additionally, the expression of vascular epidermal growth factor receptor (VEGFR), epidermal growth factor receptor (EGFR), and platelet-derived growth factor (PDGF) was decreased, proposing a potential protective effect of statins against HCC [28].

*4.5. Autophagy*

Autophagy, a crucial cellular process responsible for degrading and recycling damaged organelles and macromolecules, plays a significant role in MASLD pathophysiology [29]. Impaired autophagy contributes to lipid accumulation, oxidative stress, and hepatocyte injury, which are hallmarks of MASLD progression [30]. Dysregulated autophagy affects lipophagy (degradation of lipid droplets) and mitophagy (clearance of damaged mitochondria), exacerbating hepatocyte stress and inflammation [31]. Statins may influence autophagy by enhancing its activity, thereby improving lipid clearance and reducing oxidative stress in hepatocytes [32]. This mechanism may partially explain the observed benefits of statins in MASLD patients, such as improved liver enzyme levels and reduced hepatic steatosis. Additionally, autophagy modulation may play a role in preventing hepatic stellate cell activation, thus reducing fibrosis and disease progression. Future research is essential to explore the therapeutic potential of combining statins with agents targeting autophagy in MASLD management.

## 5. Comparison of Statins with Other Lipid-Lowering Agents in MASLD

In addition to statins, other lipid-lowering agents such as fibrates and PCSK9 inhibitors have been investigated for their potential roles in MASLD management. Fibrates, which activate peroxisome proliferator-activated receptors (PPARs), are effective in reducing triglyceride levels and improving lipid profiles [33]. However, their impact on liver-specific outcomes in MASLD remains limited [34], with some studies indicating potential hepatotoxicity in advanced liver disease.

PCSK9 inhibitors, on the other hand, work by reducing low-density lipoprotein cholesterol (LDL-C) through the inhibition of PCSK9-mediated LDL receptor degradation. While PCSK9 inhibitors show robust LDL-C-lowering effects and cardiovascular benefits, evidence regarding their efficacy in MASLD treatment is still emerging [35]. These agents are generally well tolerated, but their high cost and limited long-term data on liver-specific outcomes may restrict widespread use.

Compared to these alternatives, statins exhibit a broader range of benefits beyond lipid-lowering, including anti-inflammatory, anti-fibrotic, and antioxidant effects, making them a cornerstone in MASLD management. However, a tailored approach considering patient-specific factors and disease severity is essential when selecting between statins and other lipid-lowering agents.

## 6. Potential Synergy Between Statins and Other Therapies for MASLD

Emerging evidence suggests that combining statins with other therapeutic agents may enhance treatment outcomes in MASLD. For instance, antioxidants such as vitamin E and N-acetylcysteine have demonstrated potential in reducing oxidative stress, a key driver of liver damage in MASLD [7]. When used alongside statins, these agents could amplify the antioxidant effects, further mitigating liver inflammation and progression.

Anti-diabetic agents, particularly GLP-1 receptor agonists and SGLT2 inhibitors, also show promise in MASLD management due to their effects on weight reduction, glycemic control, and the improvement of hepatic steatosis [36,37]. Combining these agents with statins may address both metabolic and cardiovascular risks while targeting liver-specific pathologies.

Anti-inflammatory drugs, including PPAR agonists and novel agents targeting cytokine pathways, represent another avenue for synergy [38]. Statins' anti-inflammatory effects may complement these therapies, potentially reducing the progression to steatohepatitis and advanced fibrosis [39].

While these combinations hold promise, further clinical studies are needed to validate their efficacy and safety. Tailored therapeutic approaches that consider the patient's disease stage, comorbidities, and overall health status will be critical in optimizing MASLD management.

## 7. Clinical Evidence

Several systematic review and meta-analysis studies demonstrated the efficacy and safety of statin [40,41]. One meta-analysis included 14 studies with a total of 1,247,503 participants in May 2021. The analysis showed that statin use significantly reduced the risk of developing NAFLD (OR 0.69) and improved liver histology outcomes. Statins were associated with significant reductions in ALT (−27 U/I), AST (−11 U/L), and GGT (23 U/L) levels in patients with NAFLD at baseline. Furthermore, there was a notable reduction in steatosis grade, NAFLD activity score (NAS), necro-inflammatory stage, and significant fibrosis (OR 0.2). However, the effect on the fibrosis stage was not significant [42]. Another meta-analysis included 13 studies with 789 participants between 2007 and 2020. Statins showed significantly improved liver function tests including ALT (mean difference range 7–54 U/L), AST (7–38 U/L), and GGT (10–36 U/L). There was also a significant decrease in steatosis grade and NAFLD activity score (NAS) [43].

The results of this study are consistent with previous nationwide studies that have also shown the benefits of statins for both NAFLD and NASH. The Rotterdam Study, a population-based cohort from the Netherlands conducted between 2009 and 2014 enrolled 5967 participants, and the PERSONS cohort, which consists of well-characterized biopsy-proven Chinese NAFLD patients enrolled between 2016 and 2019 enrolled 569 participants, supported these findings. In both cohorts, statin use was inversely associated with NAFLD in the general population compared to the participants with untreated dyslipidemia. Furthermore, statin use was also inversely associated with NASH in NAFLD patients. The researchers suggest that adequate prescription of statins could help reduce the disease burden of NAFLD [44].

A nationwide study from the National Health Information Database of the Republic of Korea included over 11 million subjects and followed them from 2010 to 2016. The results showed that statin use was associated with a reduced risk of developing NAFLD (adjusted OR 0.66), independent of diabetes mellitus. Among the subjects with established NAFLD, statin use also reduced the risk of significant liver fibrosis (adjusted OR 0.43). These findings suggest that statins may have a beneficial effect in preventing NAFLD and slowing the progression of liver fibrosis [45].

Moreover, in this population-based cohort study utilizing the Taiwan National Health Insurance Research Database (NHIRD), the initial cohort consisted of 480,426 patients with T2DM after propensity score matching, with 137,895 in the statin user group and 137,895 in the non-statin user group. The study found that specific types of statins, such as rosuvastatin, pravastatin, atorvastatin, simvastatin, and fluvastatin, as well as higher cumulative doses, were associated with a reduced risk of decompensated liver cirrhosis in patients with T2DM. There was also a dose–response relationship, with an optimal daily intensity of statin use corresponding to the lowest risk of decompensated liver cirrhosis. The estimated daily dose recommendations for different statins based on optimal defined daily dose to reduce the risk of decompensated liver cirrhosis were as follows: Simvastatin 26 mg, Atorvastatin 18 mg, Rosuvastatin 9 mg, Pravastatin 26 mg, Pitavastatin 2 mg, Lovastatin 40 mg, and Fluvastatin 53 mg [46].

Finally, regarding cancer-related mortality, evidence from a large US prospective cohort study including 10,821 participants with NAFLD from the National Health and Nutrition Examination Survey (NHANES) showed that statin use was associated with a

43% lower risk of cancer mortality in multivariable analysis. The duration of statin use also had an impact, with statin use for 1 to 5 years decreasing cancer mortality by 35%, and statin use for over 5 years decreasing it by 56%. Furthermore, statin use was found to decrease the risk of cancer mortality in NAFLD patients with both low and high risk of liver fibrosis [47]. The review of each statin and the clinical evidence is summarized as follows:

### 7.1. Simvastatin

Limited data exist on the use of simvastatin in patients with NAFLD. A pilot randomized placebo-controlled study involving 16 patients with NASH and dyslipidemia found that although there were significant improvements in the serum lipid profile, there was no statistically significant improvement in serum aminotransferases, hepatic steatosis, necro-inflammatory activity, or stage of fibrosis after treatment with simvastatin 40 mg/day compared to placebo at a 1-year follow-up [48].

Another retrospective study involving 45 patients with NAFLD, metabolic syndrome, and increased cardiovascular risk evaluated the safety and efficacy of simvastatin monotherapy at a dose of 20 mg/day or a combination of simvastatin 10 mg/day with ezetimibe 10 mg/day. After a 6-month treatment period, both the combination therapy and simvastatin monotherapy resulted in a significant decrease in ALT and AST levels. Specifically, simvastatin monotherapy showed a dose-dependent effect, with significant reductions in ALT levels ranging from 66 to 29 U/L and AST levels ranging from 59 to 24 U/L, suggesting a potential dose-dependent effect of simvastatin. These findings demonstrate the effectiveness and safety of simvastatin therapy in patients with NAFLD [49].

### 7.2. Atorvastatin

Several studies have provided evidence supporting the positive effects of atorvastatin on liver-specific outcomes in NAFLD/NASH. In a pilot study involving 25 NAFLD patients with dyslipidemia, atorvastatin at doses ranging from 10 to 80 mg/day showed promising results. At 6 and 12 months of treatment, 36% and 20% of patients, respectively, achieved normal transaminase levels, while the remaining patients experienced a reduction in baseline levels by 10% [50].

In a phase 2 randomized placebo-controlled trial, atorvastatin at a dose of 10 mg/day effectively counteracted the increases in LDL-C levels and LDL particle concentration caused by obeticholic acid (OCA) in NASH patients [51].

Furthermore, when atorvastatin was used in combination with other agents, it also demonstrated benefits. In a study involving 1005 individuals with NAFLD, the combination of atorvastatin 20 mg/day, vitamin C, and vitamin E resulted in a reduced likelihood of hepatic steatosis development compared to placebo over an average follow-up duration of 3.6 years [52]. Another prospective randomized study involving patients with NAFLD and metabolic syndrome found that the combination of atorvastatin 20 mg/day with fenofibrate or their combination led to the normalization of liver enzymes and ultrasound findings in a significant percentage of participants, with up to 67% of the patients on atorvastatin achieving these improvements after 54 weeks of follow-up [53].

### 7.3. Rosuvastatin

Rosuvastatin has shown effectiveness in improving liver-specific endpoints in small pilot studies of NAFLD. In one pilot study involving 19 NASH patients with dyslipidemia, treatment with rosuvastatin 2.5 mg/day for 24 months did not result in significant changes in liver histology, as assessed by the non-alcoholic fatty liver disease activity score (NAS) and fibrotic stage, in all the patients. However, improvements were observed in 33.3% of the individual patients for both NAS and fibrotic stages, while stability was seen in

33.3% and 55.6%, respectively [54]. Another prospective study of 20 NASH patients with metabolic syndrome and dyslipidemia treated with rosuvastatin 10 mg/day showed the complete resolution of NASH in liver biopsy and ultrasonography assessments at 3 months of follow-up [55]. This suggests a potential dose-dependent treatment response. Similarly, in another small prospective study involving 23 NAFLD patients with dyslipidemia receiving rosuvastatin 10 mg/day, all the patients showed the normalization of liver enzymes after 8 months of treatment [56]. Additionally, in a prospective randomized study of 40 NAFLD patients with metabolic syndrome, rosuvastatin 10 mg/day treatment resulted in a decrease in intrahepatocellular lipid content as evaluated by H-MRS, a non-invasive technique, and lipid parameters were improved compared to placebo after 52 weeks of follow-up [57]. Similarly, a prospective randomized study compared the effects of rosuvastatin, metformin, and pioglitazone in NAFLD patients. The rosuvastatin group showed the greatest improvements in ultrasound scores for NAFLD, along with the most significant improvement in liver enzyme levels at 24 weeks, although none of the patients experienced liver enzyme elevation exceeding three times the upper limit of normal [58].

### 7.4. Pravastatin

Limited small studies have investigated the effect of pravastatin on NAFLD. In a small pilot prospective study involving five biopsy-proven NASH patients, pravastatin 20 mg/day led to the normalization of liver enzymes in all the patients. Additionally, variable degrees of improvement were observed in the grading of NASH, with three cases showing improvement in inflammation extent and one case showing improvement in steatosis degree. However, there was no change in the staging score of fibrosis at the 6-month follow-up [59].

In another multicenter randomized placebo-controlled study, 326 patients with dyslipidemia and known chronic liver disease, with 64% having NAFLD, were treated with high-dose pravastatin of 80 mg/day. After 36 weeks, pravastatin significantly improved the serum lipid profile, and there was no statistically significant difference in ALT elevation compared to the placebo group. These findings suggest that pravastatin is a safe option for beneficially modifying the lipid profile in NAFLD patients [60].

### 7.5. Pitavastatin

Pitavastatin has demonstrated effectiveness in reducing LDL-C levels and increasing HDL-C levels, particularly in individuals with pre-diabetes or diabetes. This favorable metabolic profile shows promise for the management and prevention of NASH [61,62]. However, the clinical evidence on the efficacy of pitavastatin in NASH is limited and yields controversial outcomes.

A randomized, placebo-controlled study of pitavastatin 4 mg/day in 50 patients with overweight and insulin resistance found no effect on endogenous glucose production or insulin-stimulated glucose uptake. Moreover, there was no change in liver fat fraction compared to placebo after a 12-week follow-up period [63].

However, in a pilot study involving 20 patients with biopsy-proven NASH and dyslipidemia, treatment with pitavastatin 2 mg/day for 12 months led to significant improvements in ALT levels. While NAS score and fibrosis stage did not change significantly in all the patients, they did improve by 54% and 42% in individual patients, respectively [64]. In another 12-week prospective, randomized study involving 189 patients with mild-to-moderate elevation of hepatic enzymes, pitavastatin 2–4 mg/day was found to reduce the severity of hepatic steatosis, as measured by nonenhanced computed tomography. This effect was particularly evident in subjects with clear hepatic steatosis at baseline [65].

### 7.6. Lovastatin

A multicenter study of 87 NAFLD patients with dyslipidemia treated with lovastatin 10 mg/day demonstrated significant reductions in liver enzymes and cholesterol levels, which were observed as early as the first 2 months and extended to 4 months of treatment. In addition, there was a decline in the AST–platelet ratio index (APRI), which is a scoring system representing liver fibrosis [66].

### 7.7. Fluvastatin

There have been no studies specifically focusing on fluvastatin in NAFLD patients [67].

Table 2 demonstrates the summary of the clinical studies on statins in NAFLD and NASH patients. This table provides an overview of the key clinical studies evaluating the effects of various statins in patients with NAFLD and NASH. It includes study designs, patient populations, dosages, treatment durations, key findings, and relevant comments to highlight the safety and efficacy of statins in managing liver and metabolic parameters.

**Table 2.** Summary of clinical studies on statins in NAFLD and NASH patients.

| Statin | Study Design and Population | Dose and Duration | Key Findings | Comments |
|---|---|---|---|---|
| **Simvastatin** | Pilot RCT, 16 NASH patients with dyslipidemia | 40 mg/day for 1 year | Improved lipid profile; no significant change in liver enzymes, steatosis, or fibrosis [48]. | Limited data; small sample size. |
| | Retrospective, 45 NAFLD patients with metabolic syndrome | 20 mg/day or 10 mg/day + ezetimibe, 6 months | Significant ALT and AST reductions; dose-dependent effects observed [49]. | Demonstrates dose-dependent safety. |
| **Atorvastatin** | Pilot, 25 NAFLD patients with dyslipidemia | 10–80 mg/day, 6–12 months | Normalized transaminase levels in 36% (6 months) and 20% (12 months) of patients [50]. | Promising liver-specific outcomes. |
| | RCT, NASH patients | 10 mg/day | Counteracted LDL-C increase caused by obeticholic acid [51]. | Effective for lipid control. |
| | Prospective, 1005 NAFLD patients | 20 mg/day + vitamin C and E, 3.6 years | Reduced hepatic steatosis risk [52]. | Benefits in combination therapy. |
| | RCT, NAFLD patients with metabolic syndrome | 20 mg/day with fenofibrate, 54 weeks | Liver enzyme normalization in 67% of patients [53]. | Effective for enzyme normalization. |
| **Rosuvastatin** | Pilot, 19 NASH patients | 2.5 mg/day for 24 months | NAS and fibrosis improvement in 33.3% of patients [54]. | Suggests a dose-dependent response. |
| | Prospective, 20 NASH patients with metabolic syndrome | 10 mg/day for 3 months | Complete resolution of NASH on biopsy/ultrasound [55]. | Promising for short-term outcomes. |
| | RCT, 40 NAFLD patients | 10 mg/day for 52 weeks | Reduced intrahepatocellular lipid content and improved lipid parameters [57]. | Non-invasive H-MRS evaluation. |
| **Pravastatin** | Pilot, 5 NASH patients | 20 mg/day for 6 months | Liver enzyme normalization in all; variable improvement in NASH inflammation/steatosis [59]. | Small study, no fibrosis improvement. |
| | RCT, 326 patients with chronic liver disease (64% NAFLD) | 80 mg/day for 36 weeks | Improved lipid profile; no significant ALT elevation compared to placebo [60]. | Safe for lipid modification. |

**Table 2.** *Cont.*

| Statin | Study Design and Population | Dose and Duration | Key Findings | Comments |
|--------|---------------------------|-------------------|--------------|----------|
| **Pitavastatin** | RCT, 50 insulin-resistant patients | 4 mg/day for 12 weeks | No significant effect on glucose metabolism or liver fat fraction [63]. | Limited evidence, controversial outcomes. |
| | Pilot, 20 biopsy-proven NASH patients | 2 mg/day for 12 months | ALT improvement; NAS and fibrosis improved in 54% and 42% of patients, respectively [64]. | Promising individual patient responses. |
| **Lovastatin** | Multicenter, 87 NAFLD patients with dyslipidemia | 10 mg/day for 4 months | Reduced liver enzymes, cholesterol, and APRI scores [66]. | Effective for short-term improvement. |
| **Fluvastatin** | No specific studies reported | — | — | No data available for NAFLD. |

## 8. Safety of Statins

Statin use in liver disease has been known to be a double-edged sword. Previously, statins were believed to cause liver toxicity and were recommended to be avoided. However, it is now understood that statins commonly cause mild elevations in serum alanine aminotransferase (ALT) levels, which can occur due to various factors. In contrast, severe liver toxicity from statins is extremely rare [68]. The incidence of statin-induced liver injury is estimated to be around 1 in 17,000 to 1.2 in 100,000 cases, typically due to idiosyncratic reactions [69]. Moreover, the incidence of acute liver failure in individuals exposed to statins is comparable to that of the general population, with a ratio of 1 in 130,000 versus 1 in 114,000, respectively [70].

The key consideration when caring for chronic liver patients who are being treated with statins is to measure liver enzymes before initiating statin therapy. Baseline liver enzyme levels provide critical information about the patient's underlying liver function. It is crucial to understand that an increase in ALT during statin treatment should not be automatically interpreted as a sign of ongoing liver disease or injury. Instead, it may be indicative of a condition known as "transaminitis", where liver enzymes leak without causing hepatotoxic effects in the absence of proven hepatotoxicity. This class effect is typically asymptomatic, reversible, and dose-related [71]. This concept of "transaminitis" may explain many of the observed ALT elevations in patients taking statins. The most common occurrence is a transient increase in ALT levels, which is typically asymptomatic and often observed within the initial 12 weeks of statin therapy (range 5–90 days) [72,73]. Several potential mechanisms of ALT elevation in statin use have been proposed in Figure 2. ALT is considered a more reliable indicator of potential hepatotoxicity compared to aspartate aminotransferase (AST), as elevated AST levels can also result from muscle injury. Additionally, elevated ALT levels should be confirmed through repeated measurements, as a single elevation is more suggestive of "transaminitis" rather than liver damage. Furthermore, clinicians should differentiate transient ALT elevations caused by statins from those due to underlying liver disease. It has been hypothesized that the lipid-lowering effect of statins might impact the structure of cellular membranes which are composed of phospholipid, leading to increased leakage of cellular enzymes. However, routine monitoring is not necessary in the absence of clinical signs or symptoms suggesting possible hepatotoxicity [74].

## Proposed mechanisms of ALT elevation in statin use

1. **Transient pharmacologic effects on cholesterol reduction in hepatocytes**
   "transaminitis without hepatotoxic effects"
2. **Statin-associated myopathy with myositis** (checking CK levels)
   Risk factors: advanced age, female sex, low BMI, untreated hypothyroidism, drug interactions through the CYP P450 pathway
3. **Coexisting MASH**
4. **Other potential causes of hepatitis**
5. **True drug-induced liver injury (DILI) from statins (idiosyncrasy).**

**Figure 2.** Several potential mechanisms of ALT elevation in statin use. CK, creatine kinase. BMI, body mass index. CYP, cytochrome. MASH, metabolic dysfunction-associated steatohepatitis.

## 9. Long-Term Safety in Advanced Liver Disease (Cirrhosis)

In patients with advanced liver disease, particularly decompensated cirrhosis, long-term statin use has shown potential benefits beyond lipid lowering. Emerging evidence suggests that statins can improve endothelial function, reduce portal hypertension, and exert anti-inflammatory and anti-fibrotic effects, which are particularly valuable in this population [75]. However, the risk–benefit balance should be carefully considered. Patients with advanced liver disease are more susceptible to complications such as rhabdomyolysis and hepatotoxicity, especially at higher statin doses or with lipophilic statins like simvastatin. Nonetheless, recent studies indicate that low-dose simvastatin, when carefully monitored, may provide a safer profile while retaining therapeutic benefits in patients with cirrhosis, particularly in decompensated cases [76].

Clinicians should closely monitor liver function in this population, particularly during the initiation phase and when adjusting doses. The potential benefits, including reduced risk of decompensation and liver-related mortality, must be weighed against the rare but serious risks of adverse events. Tailored approaches and patient-specific considerations are critical when prescribing statins for individuals with advanced liver disease.

## 10. Monitoring and Management

The 2014 Statin Liver Safety Task Force developed a comprehensive decision-making tool to guide clinicians in managing elevated liver enzymes in patients receiving statin therapy [77]. We propose a modified algorithm for patients with MASLD and statin use (Figure 3). In MASLD patients eligible for statin therapy, it is recommended to avoid a simvastatin dose exceeding 20 mg/day, particularly in cases of decompensated MASH cirrhosis due to the reported risk of rhabdomyolysis [76]. Prior to initiating statin therapy, liver enzyme levels should be measured. Routine monitoring is not necessary in the absence of clinical signs or symptoms suggesting potential hepatotoxicity.

## Approach to elevated liver enzyme in statin use

*Simvastatin should *not* exceed 20 mg/d in decompensated MASH cirrhosis

**MASLD patients with indication for statin therapy***

**Liver enzymes measurement before statin initiation**

- Routine liver enzymes testing is *NOT* necessary if no symptoms of hepatotoxicity
- Standard follow-up for MASLD based on risk stratification

**Aminotransferase < 3x ULN**

**Aminotransferase > 3x ULN**

Take medical Hx and perform PE to identify possible causes
Check CK levels to exclude the etiology of myositis

**DDx**
- Transient pharmacologic effects on cholesterol reduction in hepatocytes
- Coexisting MASH
- Other causes of hepatitis

1. **Discontinue statin** until clearer Dx is made
2. Stop other concurrent drugs potential hepatotoxicity
3. Lifestyle modification in obese patients
4. Perform a proper investigation

**Continue statin**, repeat liver enzyme after implementing lifestyle modification and addressing other causes

If initial testing does not reveal Dx and LFT does not improve despite discontinuing drugs and implementing lifestyle modification, consider liver Bx and/or MRI

**Figure 3.** The proposed modified algorithm is for patients with MASLD who are using statins. MASLD, metabolic dysfunction-associated steatotic liver disease. MASH, metabolic dysfunction-associated steatohepatitis. ULN, upper limit normal. Hx, history. PE, physical examination. CK, creatine kinase. DDx, differential diagnosis. Dx, diagnosis. LFT, liver function test. Bx, biopsy. MRI, magnetic resonance imaging.

In MASLD patients with elevated transaminase levels, regardless of the extent, a thorough medical history and physical examination should be conducted to identify the possible causes. Additionally, checking creatinine kinase (CK) levels can help exclude myositis as a potential cause. Patients can then be classified into two groups: those with transaminase elevation less than three times the upper limit of normal (ULN) and those with elevation exceeding three times the ULN [73,78].

For patients in the first group, the potential causes of transaminase elevation include transient pharmacologic effects on cholesterol reduction in hepatocytes, coexisting MASLD, or other causes of hepatitis. In such cases, it is generally safe to continue statin therapy. Regular monitoring of liver enzymes is advised, along with further investigations to determine any underlying liver injury. For patients in the second group with transaminase elevation exceeding three times the ULN, statin therapy should be discontinued along with any other potentially hepatotoxic drugs. A comprehensive evaluation should be pursued to determine the cause of liver injury.

Following an episode of statin-related drug-induced liver injury (DILI), common clinical questions arise regarding the safety of rechallenging, the possibility of switching to a different statin, or the consideration of a lower starting dose. Data addressing these questions are limited, and no definitive conclusions can be drawn [69].

## 11. Conclusions and Future Directions

In summary, the current studies provide evidence supporting the benefits of statins in various aspects of MASLD. Statin use has been shown to reduce the incidence of MASLD in high-risk populations and slow the progression of MASH by exerting anti-inflammatory, antioxidant, and anti-fibrotic effects. Statins can also effectively reduce liver enzymes in MASLD patients. Additionally, some studies suggest that statins may contribute to a reduction in cancer-related mortality. However, physicians may hesitate to prescribe statins to MASH patients due to concerns about baseline elevated liver enzymes. It is important to differentiate between true hepatotoxicity and the potential increase in liver enzymes after statin use. Lifestyle modifications remain a key component in the management of MASLD patients. Despite these advancements, several research gaps remain. Limited data exist regarding the long-term safety and efficacy of statin therapy in patients with severe MASLD, including those with advanced fibrosis or decompensated cirrhosis. Furthermore, the effects of statins on specific cellular mechanisms, such as autophagy and hepatic stellate cell activity, warrant further investigation. There is also a need for large-scale, randomized controlled trials to determine the optimal statin dosing and to address statin intolerance in complex cases. Nevertheless, large prospective cohort studies are needed to further support and promote the inclusion of statin therapy in the treatment guidelines for MASLD patients in the future.

**Funding:** Academic and Research Unit, Queen Savang Vadhana Memorial Hospital, Sriracha, Chonburi.

**Conflicts of Interest:** None declared.

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
