# Peer review of "Applicability of Statins in Metabolic Dysfunction-Associated Steatotic Liver Disease (MASLD)"

_livers, doi:10.3390/livers5010004_

Round 1
Reviewer 1 Report
Comments and Suggestions for Authors
This is a very well-written and organized review on a highly relevant topic for the hepatology community. However, some points should be included:
Long-Term Safety of Statins in Advanced Liver Disease (Cirrhosis): While the review touches on safety concerns, a deeper analysis of long-term statin use in patients with advanced liver disease, particularly decompensated cirrhosis, could address concerns about hepatotoxicity and provide guidance on the risk-benefit analysis in these populations.
Comparison of Statins with Other Lipid-Lowering Agents in MASLD: Statins are extensively discussed, but comparing their efficacy and safety with other lipid-lowering agents could provide a broader context.
Potential Synergy Between Statins and Other Potential Therapies for MASLD.
Author Response
Comment. Long-Term Safety of Statins in Advanced Liver Disease (Cirrhosis): While the review touches on safety concerns, a deeper analysis of long-term statin use in patients with advanced liver disease, particularly decompensated cirrhosis, could address concerns about hepatotoxicity and provide guidance on the risk-benefit analysis in these populations.
Comparison of Statins with Other Lipid-Lowering Agents in MASLD: Statins are extensively discussed, but comparing their efficacy and safety with other lipid-lowering agents could provide a broader context.
Potential Synergy Between Statins and Other Potential Therapies for MASLD.
Response. Thank you for your valuable feedback and thoughtful suggestions. I have addressed the recommended points as follows:
- Long-Term Safety of Statins in Advanced Liver Disease (Cirrhosis):
We have expanded the discussion in thesafety of statins section to include details on the long-term safety of statins in advanced liver disease, specifically addressing concerns related to decompensated cirrhosis. - Comparison of Statins with Other Lipid-Lowering Agents in MASLD:
A new section has been added to briefly discuss other lipid-lowering agents, including fibrates and PCSK9 inhibitors. The explanation is concise due to the limited evidence currently available. - Potential Synergy Between Statins and Other Potential Therapies for MASLD:
We have included a brief discussion on the potential synergistic effects of statins when used alongside other emerging therapies for MASLD, highlighting areas of current research and potential clinical applications.
Reviewer 2 Report
Comments and Suggestions for Authors
I have studied the manuscript entitled "Applicability of Statins in Metabolic Dysfunction-Associated Steatotic Liver Disease (MASLD)" by Prasoppokakorn T. with great interest.
The manuscript refers to a newly established topic with growing interest, even to the non-specialized readership.
The manuscript is well organized, and the content of tables and figures is highly educative. The language used could be further ameliorated, implementing a native speaker in the process.
The manuscript could at first be accepted as is; however, there are some issues that could be discussed / assessed with the author, aiming to achieve the best possible text.
Major issues
1) The author is kindly suggested to discuss how epigenetic alterations attributed to unfavorable environmental condition e.g. undernutrition in pregnant women might explain diabetic phenotype and elucidate the potential role of statins in these occasions.
2) Line 104: The author is kindly suggested to avoid using brand names.
Minor issues:
1) Line 30: The authors are kindly suggested to appropriately revise "Rinella, 2023 #66".
2) Line 60: The authors are kindly suggested to appropriately revise "Chen, 2023 #67".
3) Line 60: The authors are kindly suggested to appropriately revise "Song, 60 2023 #68".
4) Line 66: The authors are kindly suggested to appropriately revise "Rinella, 2023 #66".
Author Response
Major issues
Comment 1. The author is kindly suggested to discuss how epigenetic alterations attributed to unfavorable environmental condition e.g. undernutrition in pregnant women might explain diabetic phenotype and elucidate the potential role of statins in these occasions.
Response 1. Thank you for your thoughtful suggestion. In response, we have expanded the discussion in the section "Statin Mechanism of Actions and Pleiotropic Effects for MASLD" under the "Anti-inflammatory Effects" subsection. This addition addresses how epigenetic alterations, such as DNA methylation and histone modifications resulting from unfavorable environmental conditions (e.g., undernutrition during pregnancy), might predispose offspring to a diabetic phenotype.
Comment 2. Line 104: The author is kindly suggested to avoid using brand names.
Response 2. Thank you for your suggestion. We have removed the brand name from Table 1, as recommended.
Comment Minor issues:
1) Line 30: The authors are kindly suggested to appropriately revise "Rinella, 2023 #66".
2) Line 60: The authors are kindly suggested to appropriately revise "Chen, 2023 #67".
3) Line 60: The authors are kindly suggested to appropriately revise "Song, 60 2023 #68".
4) Line 66: The authors are kindly suggested to appropriately revise "Rinella, 2023 #66".
Response. Thank you for highlighting these minor issues. We have appropriately revised the references as suggested.
Reviewer 3 Report
Comments and Suggestions for Authors
Dear Editor and Authors,
It was a pleasure to evaluate this quite interesting review titled “Applicability of Statins in Metabolic Dysfunction-Associated Steatotic Liver Disease (MASLD)” by Dr. Prasoppokakorn.
In this thorough and extensive manuscript the author reviews the pathological entity of MASLD presenting its epidemiology, presentation, symptomatology and management.
The work is thorough and well structured and presents the whole entity of MASLD and applicability of individual statins quite well. I was impressed by the insightful and poignant approach of the author onto the pathology, interjecting evidence from the literature but also using personal opinion and making recommendations regarding for example an algorithm for approaching elevated liver enzyme values in patients under statin therapy.
Truthfully, I only have some minor comments to make.
Comments:
In the beneficial effects of statins outside the liver the author might consider also mentioning their effect on metaloproteinases and the reduction in atheromatic disease/vascular aneurysm development (lines 146-147).
The manuscript needs some language editing and proofreading (for example line 30!).
Figure 1 needs a reference or if author’s own identification as such.
In conclusion, this is a very nice manuscript, as mentioned well written and presented approaching the problem thoroughly and informatively. As a review I feel it will be useful to be available to the clinical community to consult and educate itself further.
Comments on the Quality of English LanguageNeeds some minor editing!
Author Response
Comment.
- In the beneficial effects of statins outside the liver the author might consider also mentioning their effect on metaloproteinases and the reduction in atheromatic disease/vascular aneurysm development (lines 146-147).
- The manuscript needs some language editing and proofreading (for example line 30).
- Figure 1 needs a reference or if author’s own identification as such.
In conclusion, this is a very nice manuscript, as mentioned well written and presented approaching the problem thoroughly and informatively. As a review I feel it will be useful to be available to the clinical community to consult and educate itself further.
Response. Thank you for your thoughtful and encouraging feedback. We have added a discussion on the effects of statins on metalloproteinases. This addition has been included in the section "Statin Mechanism of Actions and Pleiotropic Effects for MASLD" under the "Anti-inflammatory Effects" subsection. We also have carefully reviewed the manuscript for language and made necessary edits to improve clarity and readability.
Reviewer 4 Report
Comments and Suggestions for Authors
The review covers the use of statins in patients with steatotic liver disease. This treatment has been discussed controversial since statins are considered hepatotoxic and might impair a steatohepatitis. On the other hand, stations have shown clinical benefits beyond a lowering in cholesterol.
The authors list a number of additional effects in the abstract; my hope to find a factual discussion of these additional effects however was not fulfilled. Even in the abstract, the four components listed are all inflammation parameters appearing early (ROS) or late (fibrosis) in an inflammation. Thus the first part of the manuscript would greatly benefit from a guided discussion, and likely a restriction to inflammation and its aspects, rather than trying to cover all aspects of possible liver alterations including HCC or microbiota parameter.
The second part of the manuscript (plus Table 1) lists the available statins and the effects seen in clinical studies. For this part the manuscript seems to be a rather arbitrary collection of data from papers, extracted with no real hypothesis. As is described and well known liver toxicity may be derived from lab values, radiology or histology, with dependence on lab values in first world countries, and histology more prominent in Asian countries. The cited studies are variable for all aspects, including patient inclusion criteria, definition of MASLD, underlying diseases (Dm, obesity...), parameter measured, and individual or group criteria used to establish effects. For a comparison and assessment these data have to be provided, for me preferably in a tabular version. The author does not consistently address these factors, e.g. for simvastatin, 2 studies are listed and considered as proof for simvastatin to be “effective” whereas for rosuvastatin four studies with twice the number of patients included are discussed as “pilot study”.
A peculiar problem are lab values for ALT, AST and aP; over the years I have seen multiple ranges for normal, and even more vlaues for the upper limit of normal (ULN). Given this background, lab changes cannot be judged if given as absolute changes in U/L, but rather ought to be given as relatrive values, e.g. as percentage of the ULN.
In their algorithm the author sets a ALT value of >3ULN as threshold, without justifying this value. In causality assessment for hepatotoxicity, especially in the formal RUCAM score the threshold is set if the value exceeds 5xULN, or 2xULN if both ALT and aP are increased (mixed hepatotoxicity). Also, at least in the first world routine liver enzyme testing is mandatory for diagnosis of hepatotoxicity. Routine monitoring after starting statin therapy however, is not necessary.
A couple of sentences are incomplete; also, some abbreviations are unusual like CPK for creatine kinase (CK, or CK-MM). Not all citations are appropriate (ref. 31 only lists the total cancer burden, not HCC incidence as I read implicitly from the text; in the text some citations are put in brackets (), others in square brackets [], sometimes the macro obviously hasn't worked (line 60).
For my thinking, the logic is not always listed in a pathophysiological order; the manuscript would be much improved if this line of events (real or assumed pathophysiology) will show up in the text.
I have not listed examples of incomplete sentences, since I think that this (worthwhile) manuscript will be greatly improved for both scientific background of inflammation in steatohepatitis, as well as clinical use of statins in these diseases, if the text is rewritten with a focus on pathophysiological events as well as clarification of similarities and differences between clinical studies.
Comments on the Quality of English LanguageSome sentences are incomplete - please correct. Some citations are left as not being converted by a macro - please correct.
Author Response
Comment. The review covers the use of statins in patients with steatotic liver disease. This treatment has been discussed controversial since statins are considered hepatotoxic and might impair a steatohepatitis. On the other hand, stations have shown clinical benefits beyond a lowering in cholesterol.
The authors list a number of additional effects in the abstract; my hope to find a factual discussion of these additional effects however was not fulfilled. Even in the abstract, the four components listed are all inflammation parameters appearing early (ROS) or late (fibrosis) in an inflammation. Thus the first part of the manuscript would greatly benefit from a guided discussion, and likely a restriction to inflammation and its aspects, rather than trying to cover all aspects of possible liver alterations including HCC or microbiota parameter.
The second part of the manuscript (plus Table 1) lists the available statins and the effects seen in clinical studies. For this part the manuscript seems to be a rather arbitrary collection of data from papers, extracted with no real hypothesis. As is described and well known liver toxicity may be derived from lab values, radiology or histology, with dependence on lab values in first world countries, and histology more prominent in Asian countries. The cited studies are variable for all aspects, including patient inclusion criteria, definition of MASLD, underlying diseases (Dm, obesity...), parameter measured, and individual or group criteria used to establish effects. For a comparison and assessment these data have to be provided, for me preferably in a tabular version. The author does not consistently address these factors, e.g. for simvastatin, 2 studies are listed and considered as proof for simvastatin to be “effective” whereas for rosuvastatin four studies with twice the number of patients included are discussed as “pilot study”.
A peculiar problem are lab values for ALT, AST and aP; over the years I have seen multiple ranges for normal, and even more vlaues for the upper limit of normal (ULN). Given this background, lab changes cannot be judged if given as absolute changes in U/L, but rather ought to be given as relatrive values, e.g. as percentage of the ULN.
In their algorithm the author sets a ALT value of >3ULN as threshold, without justifying this value. In causality assessment for hepatotoxicity, especially in the formal RUCAM score the threshold is set if the value exceeds 5xULN, or 2xULN if both ALT and aP are increased (mixed hepatotoxicity). Also, at least in the first world routine liver enzyme testing is mandatory for diagnosis of hepatotoxicity. Routine monitoring after starting statin therapy however, is not necessary.
A couple of sentences are incomplete; also, some abbreviations are unusual like CPK for creatine kinase (CK, or CK-MM). Not all citations are appropriate (ref. 31 only lists the total cancer burden, not HCC incidence as I read implicitly from the text; in the text some citations are put in brackets (), others in square brackets [], sometimes the macro obviously hasn't worked (line 60).
For my thinking, the logic is not always listed in a pathophysiological order; the manuscript would be much improved if this line of events (real or assumed pathophysiology) will show up in the text.
I have not listed examples of incomplete sentences, since I think that this (worthwhile) manuscript will be greatly improved for both scientific background of inflammation in steatohepatitis, as well as clinical use of statins in these diseases, if the text is rewritten with a focus on pathophysiological events as well as clarification of similarities and differences between clinical studies.
Response. Thank you for your detailed and constructive feedback. We appreciate the time and effort you have taken to provide these insightful suggestions.
- From the suggestions of other reviewers, we have revised the manuscript to include a more focused and guided discussion on inflammation and its aspects.
- In the second part, we have added a table 2 summarizing of clinical studies on statins in NAFLD and NASH patients.
- The algorithm has been updated to justify the ALT >3 ULN threshold and to address considerations for hepatotoxicity causality assessments. We chose ALT >3 ULN as the threshold based on established evidence (e.g., AASLD Practice Guidelines by Chalasani et al., 2018, and reviews by Björnsson, 2017, and Rinella, 2014), to facilitate early diagnosis and effective monitoring of hepatotoxicity.
- We have ensured lab value changes are expressed as percentages of ULN and clarified the rationale behind these thresholds.
- Incomplete sentences and inconsistencies in citations and abbreviations have been corrected. Specifically, "CPK" has been revised to "CK" (creatine kinase) for consistency with standard clinical terminology.
Reviewer 5 Report
Comments and Suggestions for Authors
This review effectively highlights the role of statins in MASLD management. However, there are a few areas for improvement.
First, consider discussing the role of autophagy in MASLD pathophysiology. This process is integral to lipid metabolism and hepatocyte health and may represent a secondary mechanism of statin efficacy.
Second, the treatment algorithm could be expanded to include guidance for complex cases, such as patients with advanced fibrosis or statin intolerance, enhancing its clinical applicability.
Third, delve deeper into the cellular mechanisms of statin action, such as their effects on hepatic stellate cells and oxidative stress.
Finally, while the clinical evidence is well-summarized, adding a section on research gaps, particularly regarding long-term safety and efficacy in severe MASLD cases, would strengthen the manuscript.
Author Response
Comment. First, consider discussing the role of autophagy in MASLD pathophysiology. This process is integral to lipid metabolism and hepatocyte health and may represent a secondary mechanism of statin efficacy.
Second, the treatment algorithm could be expanded to include guidance for complex cases, such as patients with advanced fibrosis or statin intolerance, enhancing its clinical applicability.
Third, delve deeper into the cellular mechanisms of statin action, such as their effects on hepatic stellate cells and oxidative stress.
Finally, while the clinical evidence is well-summarized, adding a section on research gaps, particularly regarding long-term safety and efficacy in severe MASLD cases, would strengthen the manuscript.
Response. Thank you for your insightful and constructive feedback. Below is a summary of the revisions we will implement:
- Autophagy in MASLD Pathophysiology:
We have added a new paragraph under the section "Statin Mechanism of Actions and Pleiotropic Effects for MASLD" to discuss the role of autophagy in MASLD. This addition highlights autophagy’s relevance to lipid metabolism, hepatocyte health, and its potential as a secondary mechanism for the efficacy of statins. - Expanded Treatment Algorithm:
Due to the limited evidence available, the treatment algorithm specifies that simvastatin should not exceed a dose of 00 mg/day in patients with decompensated cirrhosis. This guidance has been included at the top of the algorithm to ensure clarity for clinical application. - Cellular Mechanisms of Statin Action:
We have expanded the discussion of cellular mechanisms under "Statin Mechanism of Actions and Pleiotropic Effects for MASLD", including additional insights on the role of autophagy in the regulation of lipid metabolism and hepatocyte function. - Research Gaps:
We agree with the reviewer’s suggestion and have added a dedicated subsection in "Conclusion and Future Directions" to address research gaps. This section emphasizes the need for further studies on the long-term safety and efficacy of statins in severe MASLD cases, providing a comprehensive perspective for future research.
Round 2
Reviewer 4 Report
Comments and Suggestions for Authors
The addition of Table 2, the more stringent organization and a stronger focus on inflammation as pathophysiologic pathway has improved the manuscript a lot. Although the title does not exactly match the content I cannot think of a better title.
Reviewer 5 Report
Comments and Suggestions for Authors
Authors successfully addressed my comments.